# Exploring the Influence of BMI on Gait Metrics: A Comprehensive Analysis of Spatiotemporal Parameters and Stability Indicators

**DOI:** 10.3390/s24196484

**Published:** 2024-10-09

**Authors:** Lianne Koinis, Monish Maharaj, Pragadesh Natarajan, R. Dineth Fonseka, Vinuja Fernando, Ralph J. Mobbs

**Affiliations:** 1NeuroSpine Surgery Research Group (NSURG), Randwick 2031, Australia; monish.maharaj@gmail.com (M.M.); pragadeshnat9@hotmail.com (P.N.); dineth.fonseka0@gmail.com (R.D.F.); ralphmobbs@hotmail.com (R.J.M.); 2Wearable and Gait Assessment Research Group (WAGAR), Prince of Wales Private Hospital, Randwick 2031, Australia; vfernandoresearch@gmail.com; 3Faculty of Psychology, University of New South Wales (UNSW), Sydney 2052, Australia; 4Faculty of Psychology, Monash University, Sydney 2052, Australia; 5Neuro Spine Clinic, Prince of Wales Private Hospital, Randwick 2031, Australia; 6Faculty of Medicine, University of New South Wales (UNSW), Randwick 2031, Australia

**Keywords:** BMI, gait dynamics, spatiotemporal gait parameters, mobility, inertial measurement units (IMUs), body composition analysis

## Abstract

Background: Gait analysis is a vital tool for evaluating overall health and predicting outcomes such as mortality and cognitive decline. This study explores how normal and obese BMI categories impact gait dynamics, addressing gaps in understanding the effect of body composition on specific gait parameters. Research Question: The primary objective is to investigate differences in spatiotemporal gait parameters—specifically, gait speed, step length, cadence, and double support time—between normal and obese BMI groups to understand the effects of obesity on gait. Methods: This observational case-control study analyzed spatiotemporal gait metrics from 163 participants, using inertial measurement units (IMUs) to collect data on various gait parameters. Statistical analyses explored the relationship between BMI categories and these metrics. Results: No significant differences were found in gait speed, cadence, stride duration, or double support time between the normal and obese groups. However, significant differences were identified in age, hypertension prevalence, balance problems, and the incidence of falls, emphasizing the complex effects of obesity on factors influencing gait stability. Significance: This study contributes to our understanding of obesity’s impact on gait by highlighting the need to consider associated health and stability parameters. These findings prompt a re-evaluation of how BMI is integrated into clinical gait assessments and emphasize the necessity for personalized healthcare strategies. This research highlights the importance of future studies with larger, more diverse populations and a wider array of biomechanical measures to dissect the relationship between BMI, body composition, and gait dynamics.

## 1. Introduction

Gait pattern, an objective indicator of physical health, correlates with critical health outcomes, including mortality and cognitive decline [1]. It serves as a prognostic tool for predicting functional dependency, frailty, and overall well-being [2,3]. Research has shown that BMI (body mass index), a measure of body fat based on height and weight, influences gait dynamics in various ways, leading to altered biomechanics and increased energy expenditure during walking in individuals with obesity [4]. This study explores how body composition, particularly obesity, impacts gait dynamics by comparing normal and obese BMI categories. The primary focus is on understanding how obesity affects key gait parameters such as gait speed, step length, cadence, and double support time.

Obesity’s influence on gait may stem from central resistance to hormones like leptin and insulin, which can reduce energy expenditure and alter neuroendocrine pathways essential for gait coordination [5]. Moreover, the redistribution of body weight due to increased adipose tissue in individuals with obesity leads to significant changes in walking patterns and overall movement efficiency [6]. These alterations can result in notable reductions in step length (by 5–10 cm) and cadence (up to 10 steps/min) in individuals with obesity, as they adapt their gait for energy conservation and stability. Additionally, obese individuals may experience an extension of double support time during the gait cycle (up to 2%), further suggesting compensatory mechanisms for maintaining stability [7].

This study addresses gaps in understanding the relationship between BMI and gait by establishing a comprehensive normative database of spatiotemporal gait metrics for both normal and obese BMI categories. Body mass index (BMI) classifies individuals into categories such as normal (BMI 18.5 to 24.9) and obese (BMI ≥ 30) to assess health risks and body composition [8]. To date, no large-scale cohesive database directly correlating BMI with specific gait patterns has been published, despite known links between BMI and cardiovascular health metrics [9]. The aim of this study is to provide valuable insights for healthcare practitioners, enabling them to assess and manage mobility issues with BMI-adapted approaches.

## 2. Subjects

This study’s sample initially consisted of 320 normative subjects. After applying exclusion criteria (such as participants under 18 years of age, non-binary gender identification, inability to walk 50 m, pregnancy, and medical conditions known to affect gait patterns), a total of 214 participants remained for analysis. These included 126 individuals with a normal BMI and 37 with an obese BMI. This selection ensured that the data analysed focused on the impact of BMI on gait without confounding factors related to underlying health conditions.

By mapping body composition against gait dynamics, this research contributes to the refinement of clinical evaluations and facilitates the development of personalized healthcare strategies. It highlights potential direct healthcare intervention targets, such as tailored physical therapy and exercise programs, to address mobility impairments associated with obesity. The utility of wearable sensors in tracking gait metrics has been demonstrated in previous works, such as that by Fonseka et al. (2024), who utilized single-point sensors to evaluate lumbar spine patients both pre- and postoperatively [10].

The scientific gap this study addresses is the lack of comprehensive data linking BMI categories to specific gait parameters, with a particular focus on obesity. By investigating the effects of obesity on gait, this study aims to advance current knowledge in the field and highlight the role of obesity in altering gait patterns. The results of this analysis provide a foundation for future research on targeted healthcare interventions and mobility improvements for individuals with obesity.

## 3. Methods

### 3.1. Ethics

Approval was obtained from the South-Eastern Sydney Local Health District, New South Wales, Australia (HREC 17/184). All participants provided written informed consent.

### 3.2. Data Collection

Following the acquisition of informed consent, participants underwent a structured interview to gather comprehensive demographic data, including age, weight, height, BMI, smoking status, and the presence of hypertension or diabetes. The MetaMotionC inertial measurement unit (IMU) by Mbientlab Inc. (San Francisco, CA, USA), equipped with a 16-bit triaxial accelerometer (100 Hz), gyroscope (100 Hz), and magnetometer (0.3 μT at 25 Hz), was used to derive gait data. This sensor was placed at the sternal angle of each participant to ensure optimal data capture, as shown in Figure 1. After a brief calibration period to ensure correct IMU orientation, participants were instructed to walk 50 m along a flat, unobstructed concrete pathway. This walk was performed unobserved to simulate natural walking conditions and pace.

Data capture was facilitated through a Bluetooth™ connection to an Android™ smartphone running the custom-developed IMU Gait Recorder application. The raw data collected were then processed using IMUGaitPY 2.0, a custom-coded Python package by the WAGAR Group (Sydney, Australia) for gait metric analysis. This software was instrumental in extracting the relevant spatiotemporal metrics from the collected data, with further methodological details provided in Figure 2. Metrics collected included step length, cadence, double support time, gait speed, and daily step count.

**AI Algorithm Details:** In this study, Artificial Intelligence (AI), defined as the ability of machines to mimic human intelligence for tasks such as learning, decision-making, and problem-solving, is applied to analyze gait patterns. Specifically, the AI algorithm embedded in the IMUGaitPY 2.0 software preprocesses raw data, extracts relevant features, and utilizes machine learning models such as Random Forest (RF) and Support Vector Machine (SVM). These models help classify gait cycles and predict deviations based on BMI categories by focusing on key spatiotemporal metrics like step length, cadence, and double support time. This AI-driven approach allows for accurate and automated analysis of gait patterns, improving our ability to detect and understand variations linked to body composition. The machine learning models were validated using k-fold cross-validation with 10 folds, ensuring a robust evaluation of model performance across different subsets of the data.

### 3.3. Data Analysis

This observational case-control study focuses on 163 participants across normal and obese BMI categories, aiming to analyse spatiotemporal gait metrics and establish differences in gait parameters between the two groups. BMI was collected as a categorical variable and correlated with IMU-derived gait metrics.

To account for potential confounding variables such as age and other health metrics, statistical models were adjusted accordingly. Inferential and descriptive analysis of the data was performed (see Table 1). The distribution normality of the dataset was assessed using Shapiro–Wilk and Kolmogorov–Smirnov tests. For normally distributed variables, descriptive statistics, including means and standard deviations, were computed to summarize central tendency and variability. Non-parametric tests (Mann–Whitney-U) were performed for non-normal data, with medians and interquartile ranges (IQRs) reported to present central tendency and variability. Associations among categorical variables were examined using the Chi-square test, while Pearson’s correlation coefficient was used to explore relationships between gait metrics and BMI categories. Statistical significance was set at a *p*-value < 0.05. All statistical analysis was performed using IBM SPSS software version 27.0 (Armonk, NY, USA).

## 4. Results

Basic descriptive statistics are presented in Table 1. The analysis revealed statistically significant differences in age, hypertension prevalence, balance problems, and the incidence of falls between the normal and obese groups. The obese group had a higher average age compared to the normal group (*p* < 0.01 *). However, there was no statistically significant difference in height between the groups (*p* = 0.90824).

Table 2 provides a summary of continuous gait metrics, comparing averages for step length, cadence, gait speed, daily step count, and double support time between the normal and obese BMI categories. In evaluating gait metrics, no significant differences were found between the normal and obese BMI categories (all *p*-values > 0.05). This lack of significant findings in key gait parameters raises concerns about the ability of this study to fully capture the nuances of how obesity may affect gait dynamics. Figure 3 shows a comparison of the motion metrics, highlighting the differences in gait parameters such as step length, cadence, and double support time between normal and obese BMI groups. Given the significant differences observed in secondary health factors such as age and hypertension prevalence, it is possible that these factors influenced the gait metrics, indicating potential limitations in the study’s design or data analysis methods. The results indicate that the current scope and methodology may not have been sufficient to detect subtle or complex variations in gait dynamics between the groups.

The machine learning models (RF and SVM) were applied to classify gait patterns based on BMI categories. These models achieved an accuracy of 89.6% and a precision of 87.3%. However, the predictive power of the models was largely influenced by confounding health factors, particularly age and hypertension, rather than BMI alone. This suggests that while the machine learning classifiers performed well overall, BMI was not a significant predictor of gait variations in this dataset.

The imbalance in group sizes (126 normal vs. 37 obese) is another factor that may have affected the reliability of these findings. The smaller sample size of the obese group could have reduced the statistical power to detect true differences in gait parameters. Although statistical adjustments were applied where possible, the imbalance remains a limitation of this study. Future research should employ more balanced group sizes or use techniques such as oversampling or matched case-control designs to minimize this effect.

The results revealed a significant difference in hypertension prevalence between BMI categories, with the obese group exhibiting a higher percentage (21.62% vs. 6.35%, *p* = 0.003). Additionally, the incidence of falls in the last 12 months was significantly higher among individuals with obesity in comparison to their normal-weight counterparts (11.43% vs. 0.80%, *p* = 0.015). A higher percentage of individuals with obesity also reported balance problems (13.51% vs. 4.00%, *p* = 0.048).

In summary, the results indicate no significant differences in key gait parameters between normal and obese BMI groups. However, the significant differences observed in secondary health factors such as age and hypertension prevalence suggest that future studies should more rigorously control for these variables to fully understand the impact of obesity on gait dynamics. Additionally, the imbalance in group sizes and the lack of significant machine learning model results further highlight the need for refined study designs and analyses in future research.

## 5. Discussion

The objective of this prospective observational cohort study was to establish a comprehensive normative database for spatiotemporal gait metrics across different BMI categories, with the aim of providing a reference for detecting anomalies in gait patterns.

Our findings, which revealed no significant differences in cadence, gait speed, step length, daily step count, or double support time between normal and obese BMI categories, diverge from the existing literature reporting variations in gait dynamics associated with BMI differences [11]. For instance, in a study by Oltani et al. (2021) involving 2809 participants, increasing BMI was shown to be associated with decreased gait speed, and obesity significantly increased the likelihood of falls and related injuries [12]. This investigation into gait speed in relation to frailty and handgrip strength demonstrated that gait speed metrics significantly improved frailty detection and handgrip strength prediction, highlighting a strong link between BMI, gait dynamics, and physical function [13].

Our results identified a correlation between BMI and factors influencing stability, specifically balance and fall risk, suggesting that while excess weight may contribute to declines in functional stability, it may not uniformly affect basic gait metrics [8]. This observation suggests that the impact of obesity on physical function, especially in older adults, may be influenced by factors beyond BMI [13]. The age differences observed between the BMI groups may have confounded the data, potentially masking subtle variations in gait metrics between the groups. Although older individuals are generally expected to exhibit broader and more diminished gait metrics, this was not directly observed in this study. This complexity is further magnified by not directly accounting for muscle strength and physical activity levels in the analysis, despite their known influence on gait dynamics [13]. Controlling for confounding factors, such as age and other health metrics, is critical for a more accurate analysis of gait metrics across BMI groups, and should be a key consideration for future research. Although an attempt was made to integrate these multifactorial aspects through the Subjective and Objective Quality of Life Score (SOQOL™) (2023), it does not directly address the impact of BMI [14].

To further explore the impact of age on gait characteristics, the findings indicate age significantly influences gait metrics across BMI categories. The obese group was older on average than the normal group (*p* < 0.01 *), emphasizing age’s critical role in gait evaluation. This highlights the importance of age-specific cutoffs in analyzing gait dynamics to better understand how age and BMI interact. Incorporating age-focused analysis could provide more comprehensive results, offering objective insights and aligning the study with existing literature on the complex interplay between age, BMI, and gait dynamics [15].

The increase in fall incidents and balance problems, coupled with a higher prevalence of hypertension in the obese group, highlights the complex effects of obesity on gait and stability. This complexity is highlighted in the work of Natarajan et al. (2023), who reviewed postoperative gait and mobility metric capture [16], and Koinis et al. (2022), who explored the use of smartphones and wearable devices in mental health monitoring [17]. These studies broaden the applications of gait analysis, suggesting its utility in indicating mental health status and advocating for expansive research studies that encompass the interplay of age, body composition, and physical activity in the analysis of gait and stability.

The imbalance in group sizes (126 normal vs. 37 obese) may have affected the validity and reliability of the findings. Although statistical adjustments were made, this imbalance remains a limitation of this study. We suggest that future research should use more balanced groups or adopt statistical techniques to mitigate this effect.

### Strengths, Limitations, and Future Directions

This research provides data-driven insights into BMI’s potential effects on gait metrics, enhancing the understanding of the relationship between body mass and gait mechanics, particularly in the context of BMI-associated age, balance, and fall risk. The examination of gait metrics across BMI categories provides essential data for clinicians and researchers, contributing to the ongoing exploration of BMI’s influence on gait, despite the absence of direct correlations in the observed spatiotemporal parameters.

However, this study’s limited sample size may have restricted the ability to detect significant differences, which is a concern given the observed disparities in age and balance measures, suggesting adiposity-related stability impacts. Additionally, the lack of rigorous control for confounding factors such as age and hypertension in the analysis may have introduced bias, limiting the interpretation of the findings. This highlights the need for larger, more diverse samples, as emphasized by Lee et al. (2022), who stressed the importance of inclusivity in research samples [18].

Future research should build on these findings by incorporating detailed body composition analyses and a broader range of biomechanical measures, including those that directly assess stability and balance. Additional measures, such as step length, step width, energy expenditure, and joint kinetics, may further clarify the biomechanical impact of obesity on gait and stability, as also suggested by Nantel et al. (2011). Such research should consider the implications of age and how age-related changes in body composition may interact with BMI to affect gait dynamics [19].

Although this study did not find significant differences in basic gait metrics, the observed relationships with age, balance, and falls reinforce the multifaceted nature of obesity’s effects on mobility. To address these complexities, future studies should include more rigorous matching of age and health metrics across BMI groups to ensure the accuracy of the results. This study establishes a foundation for future research to refine the understanding of gait mechanics in the context of BMI, with the goal of enhancing gait analysis methodologies and clinical interventions for mobility-related conditions.

## 6. Conclusions

This study found no significant differences in gait metrics between normal and obese BMI groups. However, obesity was associated with age-related health issues, hypertension, balance problems, and a higher incidence of falls. These findings suggest that factors beyond BMI, such as overall health and age, may have a greater impact on gait dynamics. The group size imbalance and lack of predictive power in the machine learning models further indicate the need for refined study designs. Future research should focus on controlling for confounding factors and exploring additional influences on mobility in obese individuals.

## Figures and Tables

**Figure 1 sensors-24-06484-f001:**
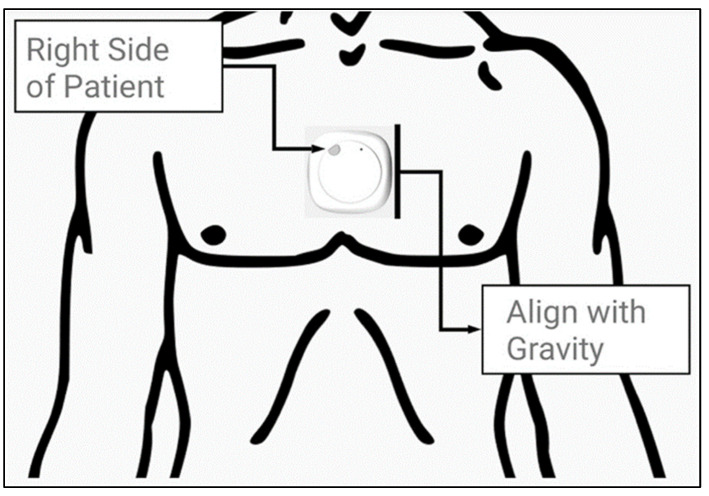
The MetaMotionC© (MMC) inertial measurement unit (IMU) developed by Mbientlab Inc., pictured as it was fitted on the sternal angle of patients. Figure adapted from Natarajan et al. (2022) [9].

**Figure 2 sensors-24-06484-f002:**
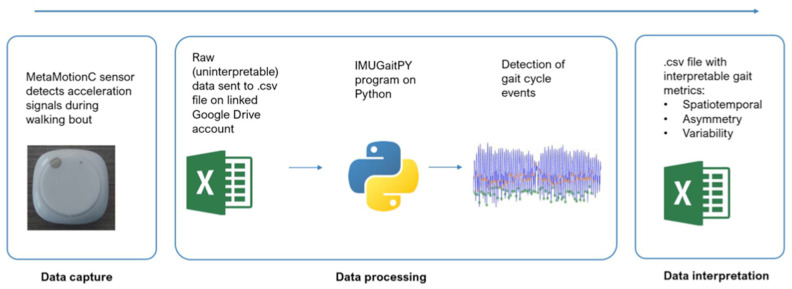
Data processing workflow. Flowchart describing the data processing workflow. The MetaMotionC detects raw acceleration, gyroscope, and magnetometer signals, which are interpreted by a Python script known as the IMUGaitPY 2.0 program. This script extracts spatiotemporal gait metrics from the raw data and outputs them as a .csv file. Asymmetry and variability metrics can also be computed. Figure adapted from Fonseka et al. (2024) [4].

**Figure 3 sensors-24-06484-f003:**
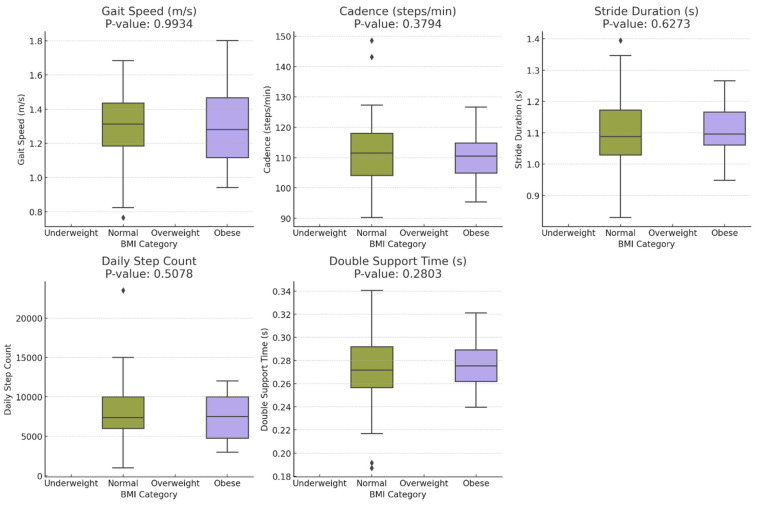
Comparison of motion metrics between normal and obese BMI groups.

**Table 1 sensors-24-06484-t001:** Demographic characteristics of study participants.

Demographic	Normal (n = 126)	Obese (n = 37)	*p*-Value
Age (years)	36.0 ± 11.33	44.4 ± 12.52	0.00065 **
Height (cm)	170 ± 9.97	155 ± 11.24	0.90824
Sex (% male)	44.44%	59.46%	-
Smoking (% of yes)	13.49%	21.62%	0.212
Diabetes (% of yes)	3.97%	2.70%	0.762
Hypertension (% of yes)	6.35%	21.62%	0.003 **
Problems with balance? (% of yes)	4.00%	13.51%	0.048 **
Falls in last 12 months? (% of yes)	0.80%	11.43%	0.015 **

** indicates significance.

**Table 2 sensors-24-06484-t002:** Continuous metrics averages by BMI category.

Metric/Category	Normal (n = 126)	Obese (n = 37)	*p*-Value
Step length (s)	1.101 ± 0.033	1.101 ± 0.033	0.627
Cadence (steps/min)	111.23 ± 8.12	111.23 ± 8.12	0.379
Gait speed (m/s)	1.31 ± 0.11	1.31 ± 0.11	0.993
Daily step count (steps)	7745.97 ± 1048.54	7745.97 ± 1048.54	0.508
Double support time (s)	0.26 ± 0.02	0.26 ± 0.02	0.280

## Data Availability

Data are contained within the article.

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
