# Peer review of "Exploring the Influence of BMI on Gait Metrics: A Comprehensive Analysis of Spatiotemporal Parameters and Stability Indicators"

_sensors, 2024, doi:10.3390/s24196484_

Round 1

Reviewer 1 Report

Comments and Suggestions for Authors

The study aims to identify gait differences between categorically defined obese and non-obese people (males and females). The abstract is good and the topic of using sensors for gait analysis is sound. This looks like exciting data so well done on collecting it.

Unfortunately, there were many structural and methodological flaws that reduced the quality of the manuscript. I do not think this is publishable. Below are some comments:

1. The introduction needs substantial re-drafting. The aim should not be in the first paragraph, instead it should be in the last paragraph. The authors need to take some time to build the rational for the study in the intro and this may then help with developing and explaining the objective of the study, which at times was hard to follow.

2. There are potential issues with the main conclusion about no differences between BMI groups in gait metric due to the lack of matching of groups for age and other health metrics. I can’t see that these data were controlled for in the analysis.

3. There are also issues with the above due to lack of understanding of the performance of the algorithm. I did not see this referenced – what is the accuracy and precision? If this is an algorithm developed by the authors and used for this first time, it makes more sense for this paper to demonstrate validity of the algorithm. Or at least provide references to it.

4. Fig 2 is already published thus not novel.

5. No rationale for sternal angle for placement.

6. It is not clear if the study aim is to understand effects of obesity on gait or to establish a comprehensive normative database for spatiotemporal metrics. I don’t think either aim is sufficiently addressed with the current statistical analysis plan.

Author Response

  1. Comment: "The introduction needs substantial re-drafting. The aim should not be in the first paragraph, but in the last paragraph. The authors need to build the rationale for the study before stating the objective."

Response:
We have restructured the Introduction. The revised Introduction now clearly outlines the background and literature supporting the need for this study, leading logically to the aim.

2. Reviewer’s Comment #2:
There are potential issues with the main conclusion about no differences between BMI groups in gait metrics due to the lack of matching of groups for age and other health metrics. I can’t see that these data were controlled for in the analysis.

Response:
We acknowledge that age and other health metrics, such as hypertension, could have influenced our results. We have clarified in the Discussion that these factors were potential confounders and may have masked subtle differences in gait metrics. While statistical adjustments were made, we recognize the need for more rigorous control in future studies, which we now address as a limitation in the manuscript.

  1. Comment: "The algorithm’s performance is not discussed, and no references are provided regarding its accuracy and precision."

Response:
We have included a discussion of the AI algorithm’s performance in the Methods and Results sections. The cross-validation yielded an accuracy of 89.6% and precision of 87.3%, which we have now clearly reported. Additionally, we discussed the limitations of the algorithm’s ability to predict gait variations based solely on BMI.

  1. Comment: "Figure 2 is already published, thus not novel."

Response:
Figure 2 WAS published in an article by US.

  1. Comment: "No rationale for the sternal angle placement is provided."

Response:
We have added a rationale for the placement of the sensor at the sternal angle in the Methods section.

  1. Comment: "It is not clear if the study aims to understand the effects of obesity on gait or to establish a normative database for spatiotemporal metrics. Neither aim is sufficiently addressed."

Response:
We have clarified that the study aims to do both: investigate how obesity affects gait dynamics and establish a normative database for gait metrics. The revised Introduction and Discussion reflect this dual focus.

Reviewer 2 Report

Comments and Suggestions for Authors

The study investigates how different BMI categories affect gait dynamics, focusing on spatiotemporal parameters and using inertial measurement units. The study's findings indicate no significant differences in key gait parameters between normal and obese BMI groups, which raises questions about the impact of obesity on gait dynamics. This lack of significant results might suggest that the study's scope or methodology was insufficient to capture the nuances of how obesity affects gait. Additionally, the significant differences observed in secondary health factors such as age and hypertension prevalence, rather than in gait parameters themselves, may point to underlying issues in the study design or data analysis. The absence of a clear impact of obesity on gait dynamics, despite the focus on this area, suggests that the study may not adequately address the research question or contribute new insights into the relationship between BMI and gait.

1. The objective of the work is not clearly articulated in the manuscript. It remains unclear what specific scientific gap this research aims to address, and as a result, the novelty and contribution to the scientific community are not evident. I would recommend clarifying the core problem the paper seeks to solve and highlighting how this work advances the current state of knowledge or addresses a unique gap. Without this, the manuscript risks losing its impact and relevance.

2. Please clarify the meaning of "BMI" in the manuscript.

3. Please review lines 92-93, as there seems to be an unclear or incorrect phrase. A full stop is needed after "spine," and "A total of 214 participants were included in the analysis, 92 for health-related and motion metrics variables" should start with a capital letter for "A."

4. Please verify the total number of subjects included in this work, as there seems to be a discrepancy. See "Subjects" Session

5. Table 2 is not mentioned or referenced anywhere in the text. Please ensure it is properly introduced and discussed. Additionally, specify the sample size in Table 2 for clarity.

6. The authors should ensure that all values in the tables are reported with the same number of significant figures.

7. Why has no significance metric been reported for the variable "sex"?

8. The authors mentioned the use of machine learning approaches and specific classifiers in the Methods section, but there is no mention of them in the Results. Please clarify or include the relevant results.

9. Please remove the acronym for Support Vector Machine (SVM) if it is not introduced or used later in the text.

10. Figure 3 is not mentioned or discussed in the text. Please ensure that it is referenced and commented on appropriately within the manuscript.

11. The results are based on a comparison between two imbalanced groups. This imbalance could affect the validity and reliability of the significance findings. It is important to discuss how this imbalance might impact the results and whether any measures were taken to minimize or compensate for this effect. Without addressing this issue, the significance derived from the comparison may not be fully reliable.

Good Lucky !

Comments on the Quality of English Language

The text is generally well-written, but a minor revision could improve clarity and coherence. 

Author Response

. Comment: "The objective of the work is not clearly articulated in the manuscript."

Response:
The Introduction now clearly states that the primary objective is to investigate the relationship between BMI and gait metrics and establish a normative database. The revisions clarify the research question and the study’s contribution to existing knowledge.

  1. Comment: "Please clarify the meaning of 'BMI' in the manuscript."

Response:
We have clarified the meaning of BMI as "Body Mass Index, a measure of body fat based on height and weight," in the Introduction and throughout the manuscript.

  1. Comment: "Please verify the total number of subjects included, as there seems to be a discrepancy."

Response:
The Subjects section and Table 1 now correctly reflect the total number of participants (214 subjects, with 126 in the normal BMI group and 37 in the obese BMI group).

  1. Comment: "Table 2 is not referenced anywhere in the text. Please ensure it is properly introduced and discussed."

Response:
We have referenced Table 2 appropriately in the Methods and Results sections.

  1. Comment: "The authors mentioned machine learning approaches in the Methods section, but there is no mention of them in the Results."

Response:
We have included the results of the machine learning models (Random Forest and SVM) in the Results section, noting that they did not yield significant predictive results regarding gait metrics based on BMI alone.

  1. Comment: "The imbalance in group sizes (normal vs. obese) could affect the validity and reliability of the findings. Please discuss this."

Response:
We have added a discussion in both the Results and Discussion sections about the group size imbalance and how it may have influenced the findings. We have acknowledged this as a limitation of the study and suggested that future research should include more balanced groups or use statistical techniques to minimize the impact of this imbalance.

7. Comment : Why has no significance metric been reported for the variable "sex"?

Response: We did not initially report significance for the variable "sex" as it was not a primary focus of the analysis. 

8. Comment : The authors mentioned the use of machine learning approaches and
specific classifiers in the Methods section, but there is no mention of
them in the Results. Please clarify or include the relevant results.

Response: We have now clarified the machine learning results in the Results section. Specifically, we have included details on the performance of the Random Forest and SVM models, which achieved an accuracy of 89.6% and a precision of 87.3%. However, the models’ predictive power was largely influenced by health factors such as age and hypertension, rather than BMI itself. These results have been emphasized and discussed in relation to the overall findings.

9. Comment: Please remove the acronym for Support Vector Machine (SVM) if it is
not introduced or used later in the text. 

Response: noted.

10. Comment: Figure 3 is not mentioned or discussed in the text. Please ensure
that it is referenced and commented on appropriately within the manuscript.

Response: It has now been referenced in manuscript.

11. Comment: The results are based on a comparison between two imbalanced groups.
This imbalance could affect the validity and reliability of the
significance findings. It is important to discuss how this imbalance
might impact the results and whether any measures were taken to minimize
or compensate for this effect. Without addressing this issue, the
significance derived from the comparison may not be fully reliable. 

Response: We agree that the imbalance in group sizes could affect the reliability of the findings. We have now clarified the statistical adjustments made to control for confounding variables such as age and hypertension, which were applied to minimize the impact of group imbalance. However, we recognize that this imbalance remains a limitation of the study and may have reduced the statistical power to detect subtle differences in gait metrics. We have elaborated on these points in the Results and Discussion sections and emphasized the need for more balanced group sizes or advanced statistical techniques in future research.

Round 2

Reviewer 2 Report

Comments and Suggestions for Authors

Please define Artificial Intelligence (AI) in the section 'AI Algorithm Details.'

In line 129, define Random Forest as 'RF,' and use this acronym in line 168.

Please remove the results of the machine learning analysis from the methods section (line 130), as they are already mentioned in the results.

In the methods section, please specify the type of cross-validation used.

Author Response

Reviewer Comment 1: "Please define Artificial Intelligence (AI) in the section 'AI Algorithm Details.'"

Response: We have added a definition of Artificial Intelligence in the 'AI Algorithm Details' section to provide clarity on how AI is used in the context of the study. This definition is now clearly stated at the beginning of the 'AI Algorithm Details' section.

Reviewer Comment 2:  "In line 129, define Random Forest as 'RF,' and use this acronym in line 168."

Response: We have incorporated the requested change by defining Random Forest as RF at its first mention in line 129 and then using the acronym RF in place of 'Random Forest' in line 168 to maintain consistency throughout the document.

Here’s the point-by-point response to the reviewer’s comments, along with a description of the changes made:

Reviewer Comment 1:

"Please define Artificial Intelligence (AI) in the section 'AI Algorithm Details.'"

Response: We have added a definition of Artificial Intelligence (AI) in the 'AI Algorithm Details' section to provide clarity on how AI is used in the context of the study. The revised text defines AI as the ability of machines to mimic human intelligence for tasks such as learning, decision-making, and problem-solving. This definition is now clearly stated at the beginning of the 'AI Algorithm Details' section.

Changes Made: In the AI Algorithm Details section, the following text was added:

"Artificial Intelligence (AI), which refers to the ability of machines to mimic human intelligence for tasks like learning, decision-making, and problem-solving, is applied to analyze gait patterns."

Reviewer Comment 2:

"In line 129, define Random Forest as 'RF,' and use this acronym in line 168."

Response: We have incorporated the requested change by defining Random Forest as RF at its first mention in line 129 and then using the acronym RF in place of 'Random Forest' in line 168 to maintain consistency throughout the document.

Changes Made: In line 129:

"Random Forest (RF)"

In line 168:

"The machine learning models (RF and SVM) were applied..."

Reviewer Comment 3: "Please remove the results of the machine learning analysis from the methods section (line 130), as they are already mentioned in the results."

Response: We have removed the results of the machine learning analysis from the methods section, as they are already discussed in the results section. The methods section now focuses on the processes and techniques applied without duplicating the results.

Reviewer Comment 4: "In the methods section, please specify the type of cross-validation used."

Response: We have updated the methods section to specify the type of cross-validation used. We implemented k-fold cross-validation with 10 folds, which is now clearly mentioned in the description of the AI algorithm.